# Cytoreduction of Residual Tumor Burden Is Decisive for Prolonged Survival in Patients with Recurrent Brain Metastases—Retrospective Analysis of 219 Patients

**DOI:** 10.3390/cancers15205067

**Published:** 2023-10-20

**Authors:** Jonas Lin, Yannik Kaiser, Benedikt Wiestler, Denise Bernhardt, Stephanie E. Combs, Claire Delbridge, Bernhard Meyer, Jens Gempt, Amir Kaywan Aftahy

**Affiliations:** 1Department of Neurosurgery, School of Medicine, Klinikum Rechts der Isar, Technical University Munich, 81675 Munich, Germany; 2Department of Neuroradiology, School of Medicine, Klinikum Rechts der Isar, Technical University Munich, 81675 Munich, Germany; 3Department of Radiation Oncology, School of Medicine, Klinikum Rechts der Isar, Technical University Munich, 81675 Munich, Germany; 4German Cancer Consortium (DKTK), Partner Site Munich, 80336 Munich, Germany; 5Department of Radiation Sciences (DRS) Helmholtz Zentrum Munich, Institute of Innovative Radiotherapy (iRT), 81675 Munich, Germany; 6Department of Neuropathology, Institute of Pathology, School of Medicine, Klinikum Rechts der Isar, Technical University Munich, 81675 Munich, Germany; 7Department of Neurosurgery, University Medical Center Hamburg-Eppendorf, 20251 Hamburg, Germany

**Keywords:** recurrent brain metastases, neurosurgery, postoperative MRI, overall survival (OS), neuro-oncology

## Abstract

**Simple Summary:**

Appropriate therapies for brain metastases are still lacking and need further research, especially in terms of recurrent brain metastases. This is why we decided to highlight the importance of complete cytoreduction by addressing the impact of postoperative tumor burden as displayed by postoperative MRI. Our findings showed that residual tumor burden is a strong prognostic factor for survival in patients with recurrent brain metastases; operated patients showed longer survival independent of age and systemic progression.

**Abstract:**

Background: Despite advances in treatment for brain metastases (BMs), the prognosis for recurrent BMs remains poor and requires further research to advance clinical management and improve patient outcomes. In particular, data addressing the impact of tumor volume and surgical resection with regard to survival remain scarce. Methods: Adult patients with recurrent BMs between December 2007 and December 2022 were analyzed. A distinction was made between operated and non-operated patients, and the residual tumor burden (RTB) was determined by using (postoperative) MRI. Survival analysis was performed and RTB cutoff values were calculated using maximally selected log-rank statistics. In addition, further analyses on systemic tumor progression and (postoperative) tumor therapy were conducted. Results: In total, 219 patients were included in the analysis. Median age was 60 years (IQR 52–69). Median preoperative tumor burden was 2.4 cm^3^ (IQR 0.8–8.3), and postoperative tumor burden was 0.5 cm^3^ (IQR 0.0–2.9). A total of 95 patients (43.4%) underwent surgery, and complete cytoreduction was achieved in 55 (25.1%) patients. Median overall survival was 6 months (IQR 2–10). Cutoff RTB in all patients was 0.12 cm^3^, showing a significant difference (*p* = 0.00029) in overall survival (OS). Multivariate analysis showed preoperative KPSS (HR 0.983, 95% CI, 0.967–0.997, *p* = 0.015), postoperative tumor burden (HR 1.03, 95% CI 1.008–1.053, *p* = 0.007), and complete vs. incomplete resection (HR 0.629, 95% CI 0.420–0.941, *p* = 0.024) as significant. Longer survival was significantly associated with surgery for recurrent BMs (*p* = 0.00097), and additional analysis demonstrated the significant effect of complete resection on survival (*p* = 0.0027). In the subgroup of patients with systemic progression, a cutoff RTB of 0.97 cm^3^ (*p* = 0.00068) was found; patients who had received surgery also showed prolonged OS (*p* = 0.036). Single systemic therapy (*p* = 0.048) and the combination of radiotherapy and systemic therapy had a significant influence on survival (*p* = 0.036). Conclusions: RTB is a strong prognostic factor for survival in patients with recurrent BMs. Operated patients with recurrent BMs showed longer survival independent of systemic progression. Maximal cytoreduction should be targeted to achieve better long-term outcomes.

## 1. Introduction

The prognosis for patients with brain metastases (BMs) has improved due to enhanced medical care and new treatment options in a multidisciplinary setting [1,2,3]. In this context, the role of surgical intervention of primary BMs has already been described [4,5,6]. Nevertheless, surgical resection increasingly fails to achieve local control [7,8,9]. Improved prognosis on the one hand and lack of intracranial tumor control on the other have contributed to the growing incidence of recurrent BMs. Through ongoing research, a variety of treatment modalities such as systemic therapy and radiotherapy have been established [10,11,12,13]. However, the treatment of recurrent brain metastases remains challenging due to sparse data on surgical resection. Previous studies have only suggested that surgery can improve outcomes [14,15,16], but there are little objective data on the impact of residual tumor volume on survival, an aspect currently only described for primary BMs [17,18].

The association between residual tumor presence, as evidenced by using postoperative magnetic resonance imaging (MRI), and an increased risk of intracranial progression is recognized [8,19]; nevertheless, early postoperative MRI imaging has not yet been established as a neurosurgical standard in the management of BMs [20].

One of the objectives of this work is to investigate the significance of residual tumor volume/burden (RTB) as assessed by using MRI imaging and its impact on survival in patients with recurrent BMs.

In addition, it was analyzed whether surgical resection in a multidisciplinary approach for the treatment of recurrent BMs provides benefits to patient outcomes and overall survival (OS).

## 2. Materials and Methods

### 2.1. Patient Population and Data Collection

A total of 231 patients with newly diagnosed recurrent BMs presented to the neurosurgical department of the Technical University of Munich between December 2007 and December 2022, and 95 (43.4%) of these cases underwent surgical treatment. A total of 219 patients fulfilled the inclusion criteria of histopathological diagnosis of a recurrent BM and MRI imaging. Twelve patients did not meet the inclusion criteria and did not receive (postoperative) MRI. A distinction was made between 124 non-operated patients (56.6%) and 95 operated patients (43.4%). The operated patient cohort was divided into the following categories: (a) Resection of recurrent BMs but with residual tumor elsewhere in the brain. (b) Resection with no residual tumor left in the brain. Patients’ medical records, including age at diagnosis, sex, tumor location, number of BMs, date of surgery, preoperative and postoperative Karnofsky Performance Status (KPS), preoperative and postoperative tumor burden, date of death, or date of last contact (in living patients), were evaluated. Data from (postoperative) systemic therapy and radiotherapy were registered and analyzed as well.

### 2.2. Surgery

The surgical approach aimed to achieve extensive tumor removal with a focus on protecting the eloquent areas of the brain. The consistent use of intraoperative neuronavigation was the standard procedure. When required, neuromonitoring and fiber tracking were used. The decision of surgical intervention was made by the multidisciplinary neuro-oncology board. Guiding factors for these decisions included (1) symptomatic lesions, (2) the presence of mass effects, (3) intratumoral hemorrhage, (4) cases with an unclear diagnosis, and (5) larger posterior fossa tumors with a potential risk of herniation and hydrocephalus.

### 2.3. Residual Tumor Burden

Volumetric measurements were obtained through early (postoperative) MRIs (within 72 h), specifically employing T1-weighted sequences with gadolinium contrast media, to identify the volume of remaining tumor remnants. An experienced neuroradiologist (BW, 11 years of experience) and neurosurgeon (AA, 7 years of experience) performed volumetric measurements. The volumes of the contrast-enhancing tumor parts were manually segmented using the Origin^®^ software (Origin^®^, Brainlab, version 3.1, Brainlab AG, Munich, Germany).

### 2.4. Statistical Analysis

Statistical analyses were performed using R version 4.1.1 (© The R Foundation, https://www.r-project.org/, accessed on 17 October 2023). Logistic regression analyses were used to detect potential risk factors for outcome changes. A difference with a probability of error less than 0.05 was considered statistically significant. Descriptive statistics with means and standard deviations or medians with interquartile ranges were obtained for demographic variables. Survival analyses were performed using Kaplan–Meier estimates for univariate analysis and the Cox regression proportional hazards model for multivariate analysis. To identify the optimal cutoff for survival curve differences, the maximally selected log-rank statistics were determined. Survival curves separated by the resulting cutoff were then compared.

### 2.5. Standard Protocol Approvals, Registrations, and Patient Consent

This study was approved by the local ethics committee (no. 5626:12) and was conducted in alignment with the ethical standards of Helsinki and later amendments [21]. The ethics committee waived the requirement for written consent.

## 3. Results

### 3.1. Patient Population

A total of 219 patients were included. The median age at diagnosis of recurrent BMs was 60.0 years (IQR 52–69). Out of the 219 patients, 121 (55.3%) were female and 98 were (44.7%) male. Median pre- and postoperative KPS was 80 (IQR 70–80). A total of 82/219 (37.4%) patients presented with a single BM, 54/219 (24.7%) with two, 37/219 (16.9%) with three, and 46/219 (21.0%) with more than three recurrent BMs. In total, 45/219 (20.5%) patients did not receive any further adjuvant therapy, 74/219 (33.8%) underwent (postoperative) radiotherapy, 20/219 (9.2%) underwent systemic therapy, and 80/219 (36.5%) received both.

### 3.2. Survival Analysis and the Influence of Residual Tumor Volume

The mean preoperative tumor burden was 7.43 cm^3^ (SD 14.92), and postoperative tumor burden (RTB) was 3.45 cm^3^ (SD 9.159) (Table 1). A total of 124/219 (56.6%) patients were not operated on. A total of 40/219 (18.3%) patients had one or more BMs resected, but there was residual tumor in one or more sites. Complete cytoreduction was achieved in 55/219 (25.1%) patients. In 27/40 (67.5%) patients who underwent surgery, the targeted metastasis was completely resected, but residual tumor was still present at other sites in the brain. A total of 56/95 (59.0%) patients with a single recurrent BM underwent surgery, 19/95 (20.0%) with two, 10/95 (10.5%) with three, and 10/95 (10.5%) with more than three BMs. Complete resection was achieved in 48/55 (87.3%) cases with one BM, in 6/55 (10.9%) with two, and in 1/55 (1.8%) with three BMs (Table 2).

The mean postoperative tumor volume of the targeted BM was 0.34 cm^3^ (SD 1.91; range 0.00–15.60 cm^3^). Median overall survival was 6 months (IQR 2–10) (Figure 1A). A significant cutoff value for an RTB of 0.12 cm^3^ (*p* = 0.00029) was found by using maximally selected log-rank statistics for all patients, regardless of the number of BMs (Figure 1B).

Multivariate analysis showed preoperative KPSS (HR 0.983, 95% CI, 0.967–0.997, *p* = 0.015), postoperative tumor burden in cm^3^ (HR 1.03, 95% CI 1.008–1.053, *p* = 0.007), and complete vs. incomplete resection (HR 0.629, 95% CI 0.420–0.941, *p* = 0.024) to be significant (Table 3).

Comparing operated and non-operated patient collectives, longer survival was significantly associated with surgery for recurrent BMs (*p* = 0.00097) (Figure 2A). Additional analysis showed the significant impact of complete vs. incomplete resection on OS (*p* = 0.0027) (Figure 2B).

A subgroup analysis was performed to differentiate between patients with and without systemic progression; 132/219 (60.3%) patients presented with systemic progression. In this subgroup, a cutoff RTB of 0.97 cm^3^ (*p* = 0.00068) was identified (Figure 3A); patients who underwent surgery again showed prolonged OS (*p* = 0.036) (Figure 3B).

### 3.3. Impact of Adjuvant Therapy on Survival

Regarding the different therapeutic regimens, only systemic therapy after relapse (*p* = 0.048) had a significant effect on survival (Figure 4A). However, combination therapy was associated with a better outcome than single therapy (*p* = 0.036) (Figure 4B).

## 4. Discussion

### 4.1. Prognostic Factors for Survival and Influence of Residual Tumor Volume

The extent of resection (EOR), initial functional status (KPS), and progression-free survival (PFS) are key factors that influence median OS for recurrent BMs [15,22]. In the present study, median OS was 6 months (IQR 2–10). When matching these prognostic factors, preoperative KPSS (HR 0.983, 95% CI, 0.967–0.997, *p* = 0.015), complete resection (HR 0.629, 95% CI 0.420–0.941, *p* = 0.024), and postoperative tumor burden (HR 1.03, 95% CI 1.008–1.053, *p* = 0.007) had a significant impact on OS. These results highlight the relevance of comprehensive cytoreduction, which is confirmed by a significant cutoff for a residual tumor volume (RTB) of 0.12 cm^3^ (*p* = 0.00029).

Prior studies have suggested that patients who face recurrent BMs alongside uncontrolled systemic disease may be less likely to benefit from surgery [14,16,23].

However, the benefits of surgical resection are particularly apparent in patients with symptomatic recurrent BMs. As previously mentioned, surgery results in improved functional outcomes and prolonged functional independence [11,24]. Furthermore, it should be added that median OS after neurosurgical resection of recurrent BMs is significantly influenced by baseline functional status, as already described by the prognostic factors mentioned above.

When comparing the groups of patients who underwent surgery and those who did not, it is evident that surgical intervention in the recurrence setting has an essential influence on survival (*p* = 0.00097). Regarding surgical intervention, it can further be added that complete resection provides a significant survival advantage (*p* = 0.0027). Recently, the extent of surgical resection has been shown to play a crucial role in the local progression rate of brain metastases, underscoring what has just been highlighted. Extensive surgical resection and evidence of residual tumor demonstrate a significant correlation with the rate of cerebral progression [7]. In this context, it is important to consider the efforts made in medical technology, particularly the availability and advancement of high-resolution microscopes [25,26,27] and the application of molecular biology methods, such as the use of fluorescence-guided resection [4,28,29]. These technologies have helped to improve the capabilities of BM resection.

### 4.2. Impact of Systemic Tumor Progression

In addition, the aspect of systemic progression should be considered. Most patients (60.3%) showed systemic progression, which included both recurrent BMs and the occurrence of at least one extracranial spread. According to recent research, it is controversial whether patients with advanced oncological disease with (extra-)cranial metastases would still benefit from further surgical intervention [4,10,23]. For the subgroup with systemic progression, a critical cutoff RTB of 0.97 cm^3^ (*p* = 0.00068) was again determined by using log-rank statistics, resulting in significant differences in the survival curves. Among the 132 patients with systemic progression, those who underwent surgery showed significantly prolonged OS, consistent with the results of the overall population (*p* = 0.036). These results reinforce the consensus that a progressive lesion in a patient demonstrating systemic disease and good functional status (Karnofsky performance status [KPS] > 60) presents a favorable profile for undergoing surgical intervention [7,11,15,30]. An infiltrative growth pattern observed in cerebral metastases could contribute as a potential reason for their elevated rate of local recurrence, thereby influencing individual OS [31,32]. For this reason, it is essential to verify the extent of resection (EOR) using postoperative MRI, as complete resection cannot always be justified through intraoperative estimates, as demonstrated in previous reports [8,19]. Thus, residual tumor is a well-established risk factor for local metastatic recurrences, although further investigation is needed to accurately assess the oncologic consequences of unintentionally incomplete metastatic resections [7,31].

### 4.3. Further Treatment and Outlook

An isolated surgical approach may be effective locally, but it neglects potentially present microscopic residuals and systemic metastases [31]. Radiotherapy and systemic therapy represent additional cornerstones within the interdisciplinary treatment for recurrent BMs and are proving essential not only for minimizing the local recurrence risk but also to more effectively control the overall disease [11,23,33].

In prospective randomized trials of the treatment of primary BMs, analyses have shown that WBRT and stereotactic radiosurgery of the cavity significantly improve local control compared with resection alone [34,35,36]. Studies investigating postoperative irradiation of BMs have also identified EOR as an important prognostic factor for OS [37,38]. However, our study showed that in patients who have already undergone radiotherapy and are in a recurrent situation, additional single radiotherapy does not provide a measurable prolongation of survival (*p* = 0.36).

Interestingly, another therapeutic approach, single systemic therapy, has been shown to be effective in this context (*p* = 0.048). Controlling the systemic burden of disease can help slow or stop the growth of BMs [12,33]. As this study specifically addresses the question of the benefits of surgical cytoreduction, two aspects can be highlighted. First, re-evaluating tumor molecular biology could lead to novel systemic treatment strategies, and second, resecting symptomatic recurrent BMs might restore clinical condition, enhancing the chance of subsequent adjuvant treatment and consequently increasing OS, as observed [14,24]. Taking into consideration the patient cohort without additional therapy, which presents inferior survival in comparison, reinforces this assertion.

Furthermore, the combined use of radio and systemic therapy was found to be superior to single therapeutic approaches (*p* = 0.036). Radiation therapy and systemic treatment can generate synergistic effects in terms of both local and systemic control, helping to control and eliminate the residual tumor and micrometastases [2,39,40]. Advances in combining surgical resection, radiotherapy, and systemic therapy have the potential to improve the quality of life and survival of patients with BMs. These patient-centered therapeutic approaches benefit from the close interdisciplinary collaborations between neurosurgeons, radiation oncologists, and medical oncologists that are now considered an established standard of care [1,10,23].

In this context, close interdisciplinary cooperation becomes an integral part of modern neuro-oncology practice, contributing significantly to the further development and improvement of patient care. However, in the setting of BM recurrence, the available data are limited, and further research is needed.

In summary, the present study illustrates that RTB has a significant impact on OS in cases of recurrence and particularly emphasizes the importance of performing cytoreductive surgery as comprehensively as possible. Thus, findings that have already been observed for first-time BMs [17] could also be confirmed for recurrent brain metastases.

### 4.4. Study Limitations

The retrospective study design at a single center limits the results of our analysis. The subsequent selection of patients, as well as varying treatment depending on functional status, has resulted in a heterogenous patient cohort. In addition, these patients also have differences in the type of primary tumors and tumor stages. Only patients who had MRI follow-up examinations were included. Special attention was paid to patients who were in a favorable oncologic condition, whereas CT imaging was not included for patients who were unable to obtain MRI imaging.

Additionally, the number of patients included in this current study is limited. A more extensive population would be required to obtain results of higher significance. The patient population was also diverse with respect to different adjuvant and systemic therapeutic approaches prior to the diagnosis of recurrent metastases. Adjuvant therapy plans after initial and repeat craniotomy are still a matter of debate and need to be further explored in future research [34,41,42,43].

Since a relatively substantial proportion (60.3%) of patients had an uncontrolled systemic health status, and metastatic tumor burden in the brain can be considered an expression of systemic disease itself, it is recommended that future research analyze the potential impact of systemic disease status on the rate of local recurrences [7].

## 5. Conclusions

RTB is a strong prognostic factor for survival in patients with recurrent brain metastases. The routine identification of residual tumors through conventional (postoperative) MRI imaging as well as the advancement of more aggressive surgical methods present significant neuro-oncological hurdles. The goal should always be to achieve maximal cytoreduction, even in recurrent BMs, regardless of age and systemic progression.

Addressing these challenges holds the potential to enhance local tumor control and consequently improve patient outcomes.

## Figures and Tables

**Figure 1 cancers-15-05067-f001:**
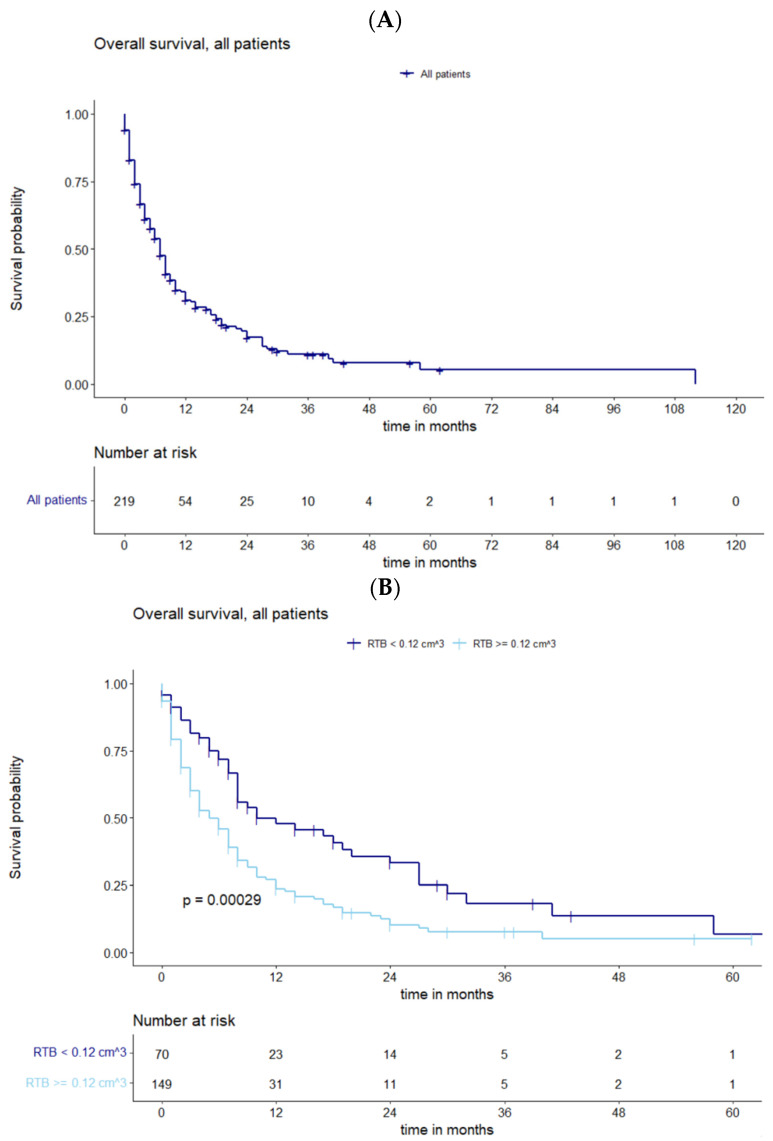
(**A**) Overall survival of all patients with recurrent BMs. (**B**) Survival analysis in all patients for subgroups of cutoff residual tumor showing statistically significant deviations in survival curves.

**Figure 2 cancers-15-05067-f002:**
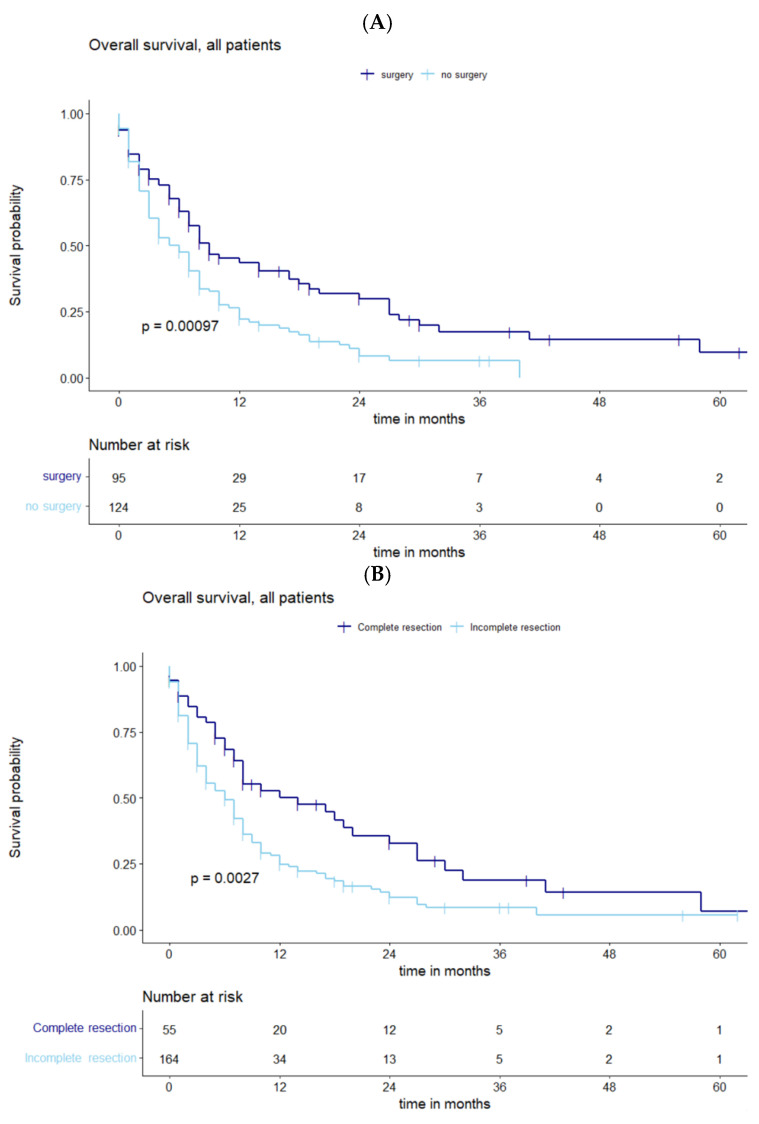
(**A**) Kaplan–Meier curves representing OS of all patients with recurrent BMs stratified by surgery vs. no surgery, showing a statistically significant correlation. (**B**) Functions of OS demonstrating significantly divergent survival curves after complete vs. incomplete resection.

**Figure 3 cancers-15-05067-f003:**
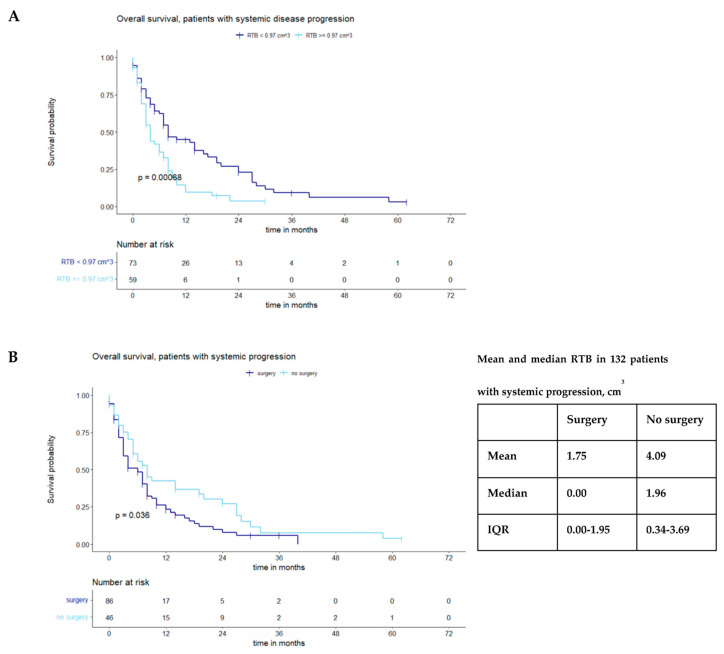
(**A**) Maximally selected log-rank statistics displaying the cutoff of postoperative tumor volume in a subgroup with systemic tumor progression. (**B**) Divergence in survival in this subgroup stratified by surgery vs. no surgery.

**Figure 4 cancers-15-05067-f004:**
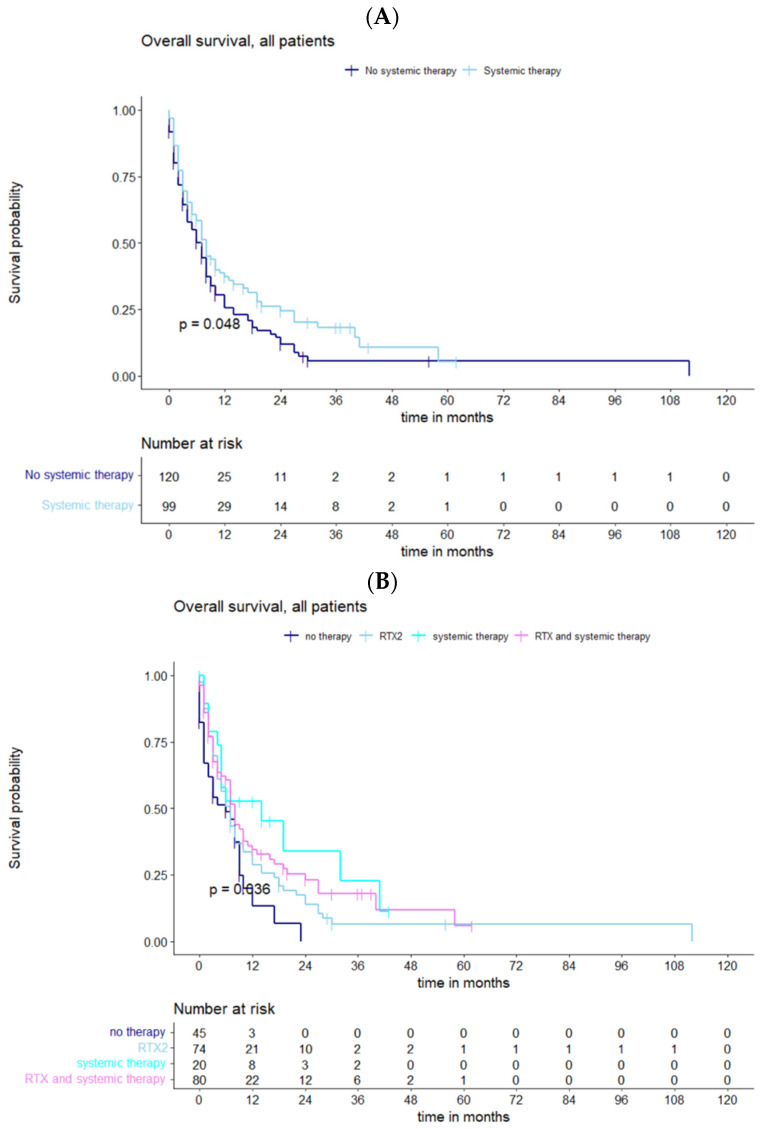
(**A**) Functions of OS demonstrating significantly divergent survival curves after systemic vs. no systemic therapy. (**B**) Survival estimates in subgroups which underwent different regimes of therapy demonstrating significantly divergent survival curves in favor of combined therapy modalities.

**Table 1 cancers-15-05067-t001:** Demographics, tumor data, and adjuvant therapy.

Demographics, N (%) or Median (Range/IQR).	
Sex	F 121/219 (55.3) M 98/219 (44.7)
Age	60.0 (IQR 52–69)
Karnofsky Performance Status Scale (KPSS)	
Preoperative KPSS	80% (IQR 70–80)
Postoperative KPSS	80% (IQR 70–80)
Number of metastases, N (%)	
1	82/219 (37.4)
2	54/219 (24.7)
3	37/219 (16.9)
>3	46/219 (21.0)
Adjuvant therapy, N (%)	
No therapy	45/219 (20.5)
Radiotherapy	74/219 (33.8)
Systemic therapy	20/219 (9.2)
Combined	80/219 (36.5)
Tumor burden (cm^3^)	
Preoperative	
Median	2.4 cm^3^ (0.8–8.3 cm^3^)
Range	0.02–184.33 cm^3^
Mean	7.43 cm^3^ (SD 14.92)
Postoperative	
Median	0.5 cm^3^ (0.0–2.9 cm^3^)
Range	0.00–98.34 cm^3^
Mean	3.45 cm^3^ (SD 9.159)

**Table 2 cancers-15-05067-t002:** Management and surgical outcomes in recurrent BMs.

Recurrent Brain Metastasis Management, N (%)	
No resection of BMs	124/219 (56.6)
Surgery with RTB left at other locations	40/219 (18.3)
Surgery with no RTB left Complete resection	55/219 (25.1)
Resection Outcomes with Targeted Recurrent BMs, N (%)	
Complete resection of targeted BMs with RTB left at other locations	27/40 (67.5)
Incomplete resection of targeted BMs with RTB left at other locations	13/40 (32.5)
Surgery Based on the Number of BMs, N (%)	
1	56/95 (59.0)
2	19/95 (20.0)
3	10/95 (10.5)
>3	10/95 (10.5)
Complete Resection Based on the Number of BMs, N (%)	
1	48/55 (87.3)
2	6/55 (10.9)
3	1/55 (1.8)
>3	0/55 (0.0)

**Table 3 cancers-15-05067-t003:** Multivariate Cox regression analysis.

Multivariate Analysis Parameters	Hazard Ratio (IQR)	*p*-Value
Sex	0.8046 (0.5824–1.1117)	0.18753
Age (year)	1.002092 (0.9889–1.0155)	0.75745
KPSS at admission	0.982465 (0.9686–0.9965)	0.01466
Preoperative tumor volume (cm^3^)	0.9966 (0.9828–1.0106)	0.63142
RTB (cm^3^)	1.030376 (1.0081–1.0532)	0.00736
RTB = 0	0.628666 (0.4202–0.9405)	0.02391

## Data Availability

The data that support the findings of this study are available from the corresponding author, [A.K.A], upon reasonable request.

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
