# Peer review of "Cytoreduction of Residual Tumor Burden Is Decisive for Prolonged Survival in Patients with Recurrent Brain Metastases—Retrospective Analysis of 219 Patients"

_cancers, 2023, doi:10.3390/cancers15205067_

Round 1
Reviewer 1 Report
Comments and Suggestions for Authors
The manuscript is well written. Small corrections in the English language would be useful. The structure of the manuscript is good. The statement is made. It would make sense to go into more detail about the combined surgical, radio-oncological therapy of metastases, as this is now a standard of care in every neurosurgical oncological center. The many statistics should be shortened as they can easily confuse the reader (eg. Figure 6). On the discussion, one should also mention the technical effords (better high-resolution microscopes) and molecular biological possibilities (use of 5-ALA-experimental) for better resection options for brain metastasis with references to the literature.
Comments on the Quality of English Languages.above.
Author Response
Authors’ Response: Thank you very much for your review and comments.
We have restated the section addressed in light of your comments. When the extend of resection was brought up in the discussion, it was appropriate to refer to the efforts of medical technology as you mention them and in the outlook of the discussion, we have tried to go into more detail about the interdisciplinary treatment of brain metastases. We have shortened the statistics to provide clarity. Thank you.
Changes: Additions were made (marked).

Reviewer 2 Report
Comments and Suggestions for Authors
General comment:
The main result of this retrospective analysis seems to be that in patients with intracranial relapse of brain metastases, overall survival is determined by the postoperative residual tumor volume as assessed with MRI. As detailed below, the authors should present a hypothesis, give clear inclusion criteria and restrict the types and number of analyses they present to the hypothesis under question.
Specific comments:
Introduction:
line 53: lack of intracranial tumor control
line 65: as assessed by MRI imaging
line 69: patient outcome other than survival was not analysed here
Methods:
line 73: 'patient collection' sounds strange, use e.g. patient cohort
line 74: please clearly indicate the inclusion criteria
line 80: within the same paragraph, you state that 94 opposed to 95 patients had resection
line 82: KPSS is the scale, the patients have a KPS
line 98: why where only lesions < 10mm in one dimension regarded as residual tumor ?
line 102 - 105: These definitions do not make much sense to me. I would suggest to analyse the following categories:
A) No resection of metastases
B) Resection of one or more metastases, but residual tumor left at one or more locations
C) Resection, no residual tumor left anywhere in the brain
The term 'complete resection' does not make sense to me if one metastasis is completely resected while others are still in place. Therefore, please state how many patients with complete resection of the targeted metastases had residual tumor at other sites after surgery, and also provide the proportion of patients which 1,2,3, >3 mets that had surgery/complete resection
Results:
line 136: please also state range, mean and standard variation of volumes
Figures: I would really recommend to restrict the number of survival curves to those relevant for the main hypothesis of the paper and to those with significant result.
Discussion: Much too long, please focus more on your main results
Some spelling mistakes need to be corrected and some grammatical inaccuracies need to be improved.
Author Response
Authors’ Response: Thank you very much!
Of course, we will review the manuscript, change the sections according to your review and will add some more data!
Introduction and Methods: we have added your suggestions for improvement. As for the unclear definition, we have taken your suggestion and added a table with the data you requested. We hope this provides clarity. On the question of why only lesions < 10 mm in one dimension are considered residual tumor, this is an erroneous statement and should therefore not be considered further.
Results: We have limited the many statistics to the relevant and significant results for our study.
Discussion: We have limited the discussion to the aspects mentioned above. Thank you very much!
Changes: Additions were made and marked.

Round 2
Reviewer 2 Report
Comments and Suggestions for Authors
Thank you for revising the paper which has now clearly improved. On the whole, my suggestions have been applied.
Comments on the Quality of English LanguageSome minor editing needed